# Baseline levels and longitudinal changes in plasma Aβ42/40 among Black and white individuals

Chengjie Xiong[1,2,10], Jingqin Luo [3,4,10], David A. Wolk[5], Leslie M. Shaw[5,6], Erik D. Roberson [7], Charles F. Murchison[7], Rachel L. Henson[2], Tammie L. S. Benzinger [2,8], Quoc Bui[1], Folasade Agboola[1], Elizabeth Grant[1], Emily N. Gremminger[1], Krista L. Moulder[2], David S. Geldmacher[7], Olivio J. Clay[7,9], Ganesh Babulal[2], Carlos Cruchaga [2], David M. Holtzman [2], Randall J. Bateman [2], John C. Morris[2] & Suzanne E. Schindler [2] ✉

Blood-based biomarkers of Alzheimer disease (AD) may facilitate testing of historically under-represented groups. The Study of Race to Understand Alzheimer Biomarkers (SORTOUT-AB) is a multi-center longitudinal study to compare AD biomarkers in participants who identify their race as either Black or white. Plasma samples from 324 Black and 1,547 white participants underwent analysis with C₂N Diagnostics' PrecivityAD test for Aβ42 and Aβ40. Compared to white individuals, Black individuals had higher average plasma Aβ42/40 levels at baseline, consistent with a lower average level of amyloid pathology. Interestingly, this difference resulted from lower average levels of plasma Aβ40 in Black participants. Despite the differences, Black and white individuals had similar longitudinal rates of change in Aβ42/40, consistent with a similar rate of amyloid accumulation. Our results agree with multiple recent studies demonstrating a lower prevalence of amyloid pathology in Black individuals, and additionally suggest that amyloid accumulates consistently across both groups.

Biomarkers of Alzheimer's disease (AD), including fluid and imaging biomarkers of amyloid and tau pathology, have enabled a better understanding of AD pathophysiology, facilitated clinical trials that have led to the development of amyloid-lowering treatments, and increased the accuracy of clinical dementia diagnosis[1]. While cerebrospinal fluid (CSF)- and positron emission tomography (PET)-based biomarkers accurately detect AD brain pathology, the scale of testing with these modalities is limited by their requirements for specialized personnel and equipment, perceived risks, and high costs[1–4]. Additionally, individuals from minoritized groups may be less likely to present to memory clinics that perform biomarker testing with CSF and PET[5]. In contrast, blood tests are considered highly accessible, acceptable, and scalable, making blood-based biomarkers ideal tools for research, clinical trials, and clinical practice[6,7]. Blood-based biomarkers may enable testing of individuals who would not be comfortable with CSF or PET testing and may allow for testing in a community-

[1]Division of Biostatistics, Washington University, St. Louis, MO, USA. [2]Department of Neurology, Washington University School of Medicine, St. Louis, MO, USA. [3]Division of Public Health Sciences, Department of Surgery, Washington University School of Medicine, St. Louis, MO, USA. [4]Siteman Cancer Center Biostatistics and Qualitative Research Shared Resource, Washington University School of Medicine, St. Louis, MO, USA. [5]Perelman School of Medicine, University of Pennsylvania, Philadelphia, PA, USA. [6]Department of Pathology and Laboratory Medicine, University of Pennsylvania, Philadelphia, PA, USA. [7]Alzheimer's Disease Center, Department of Neurology, University of Alabama at Birmingham, Birmingham, AL, USA. [8]Mallinckrodt Institute of Radiology, Washington University, St. Louis, MO, USA. [9]Department of Psychology, University of Alabama at Birmingham, Birmingham, AL, USA. [10]These authors contributed equally: Chengjie Xiong, Jingqin Luo. ✉e-mail: schindler.s.e@wustl.edu

based setting rather than a major medical center[8]. Therefore, blood-based biomarkers of AD may increase biomarker testing of individuals from minoritized groups that have historically been under-represented in AD research studies and clinical trials[8–11].

Multiple epidemiological studies have reported a higher pre-valence of dementia in self-identified Black or African American and Hispanic individuals as compared to non-Hispanic white individuals[12–14]. Despite the higher reported prevalence of dementia, several research studies have reported a lower rate of AD biomarker abnormalities in Black and Hispanic individuals compared to non-Hispanic white individuals[15–21], although other studies have found the opposite result or no differences between these groups[22,23]. The seeming disconnect between the reported prevalence of dementia and the rate of AD biomarker abnormalities has raised concerns that the major etiologies of dementia may vary across racial and ethnic groups and/or that biomarkers may not reflect AD pathology consistently across groups[19,20,24–26]. Adding to these concerns, concentrations of some plasma biomarkers, including amyloid-β 42 and 40 (Aβ42 and Aβ40, respectively) and tau phosphorylated at positions 181 and 217 (p-tau181 and p-tau217, respectively) can be affected by medical conditions (e.g., chronic kidney disease and obesity) that are more prevalent in some racial and ethnic groups[16,27–29]. However, some evidence indicates that plasma biomarker ratios such as Aβ42/40 and the ratio of phosphorylated to non-phosphorylated tau (p-tau ratio) may normalize for non-AD-related individual differences and provide more consistent performance in classifying amyloid status across groups[16,28–30].

We have previously reported that plasma Aβ42/40, as measured by a high-precision mass spectrometry-based assay, has more consistent performance in classifying amyloid status across racial groups as compared to concentrations of phosphorylated tau[16]. This finding suggests that plasma Aβ42/40, as measured by high-precision assays, may enable consistent classification of amyloid status in diverse groups. Our study and the few other studies that have compared plasma biomarkers in different racial groups only reported cross-sectional data[16,23,31–34]. Therefore, it is unknown whether the long-itudinal rates of change in plasma biomarkers vary by race. The rate of change is particularly important in clinical trials, as it represents the placebo trajectory that is intended to be modified by treatments.

In this study, we assembled a large cohort to examine plasma Aβ measures (Aβ42, Aβ40, and Aβ42/40) for potential differences in base-line levels and rates of change in self-identified Black and white indi-viduals. Participants from three AD Research Centers (Washington University, University of Pennsylvania, and University of Alabama at Birmingham) were included. Plasma samples were analyzed with the C₂N Diagnostics mass spectrometry-based assays that are currently being used in clinical trials and clinical practice[35,36]. Linear models were used to estimate the baseline levels and rates of change for plasma biomarker measures in both groups. Analyses also examined whether factors such as sex, APOE ε4 carrier status, cognitive status, or medical conditions (hypertension and diabetes) modified potential racial differences.

## Results
### Participant characteristics
The study cohort included a total of 324 Black participants and 1547 white participants with plasma Aβ measures from at least one sample (Table 1). The time interval between clinical assessment and plasma collection was 0.16 ± 0.18 years (mean ± standard deviation). Black and white participants had similar ages at baseline (70.2 ± 8.6 versus 70.5 ± 9.5 years, respectively, p = 0.26), and there was no difference in the proportion carrying an APOE ε4 allele (45.1% versus 42.6%, p = 0.35). Most participants completed at least 12 years of education, with Black participants completing slightly fewer years of education on average compared to white participants (15.3 ± 2.9 years versus 15.8 ± 2.8 years, p = 0.002). Black participants were more likely to be cognitively unimpaired than white participants at baseline (72.2%

versus 66.4%, p = 0.041) and were more likely than white participants to be female (72.2% versus 53.5%, p < 0.0001). Black participants were more likely to be overweight or obese than white participants (69.4% versus 52.0%, p < 0.0001) and were more likely than white participants to have a history of hypertension (65.4% versus 42.3%, p < 0.0001) or diabetes (17.3% versus 6.0%, p < 0.0001). A lower percentage of Black participants were fasting at the time of the baseline plasma collection compared to white participants (36.1% versus 65.5%, p < 0.0001). This is because fasting samples were collected at the time of lumbar puncture for many participants, and Black individuals were less likely to choose to undergo lumbar puncture.

### Racial differences in baseline biomarker levels
Baseline levels of plasma Aβ were compared in Black and white parti-cipants. The largest comparison available was for unadjusted plasma Aβ levels because some individuals were missing data on covariates. Unadjusted levels of plasma Aβ42/40 were higher in 324 Black indivi-duals compared to 1547 white individuals in both cognitively unim-paired (0.107 ± 0.0119 versus 0.102 ± 0.0106, p = 0.0367) and cognitively impaired groups (0.0995 ± 0.0121 versus 0.0964 ± 0.0105, p < 0.0001), consistent with Black individuals having less brain amy-loid (Fig. 1). Next, baseline levels of plasma Aβ were examined after adjustment for covariates (age, sex, APOE ε4 carrier status, years of education, cognitive status, fasting status, BMI, and status for hyper-tension, diabetes and stroke) (Table 2). Consistent with the unadjusted plasma Aβ42/40 results, the covariate-adjusted mean plasma Aβ42/40 values were higher (less abnormal) in 214 Black individuals than in 1113 white individuals (0.1201 ± 0.0030 versus 0.1155 ± 0.0030, p < 0.0001). Interestingly, the higher plasma Aβ42/40 values in Black individuals compared to white individuals resulted from lower plasma Aβ40 levels (177.8 ± 22.7 versus 199.5 ± 22.4 pg/mL, p = 0.0002); plasma Aβ42 levels were not different in Black and white individuals (26.25 ± 2.40 pg/mL versus 26.41 ± 2.36 pg/mL, p = 0.80).

Further analyses examined whether racial differences in the adjusted mean levels of plasma Aβ were modified by amyloid status, age, sex, APOE ε4 carrier status, years of education, cognitive status, BMI, hypertension, or diabetes. Amyloid status by CSF or amyloid PET did not significantly affect racial differences in plasma Aβ42/40, Aβ42, or Aβ40 (Supplementary Table 1). Racial differences in plasma Aβ42/40 were larger in cognitively unimpaired compared to cognitively impaired individuals (p = 0.017) (Supplementary Table 2), but racial differences in plasma Aβ42 or Aβ40 were not significantly affected by cognitive status (Supplementary Tables 3, 4). Racial differences in plasma Aβ42 or Aβ40 were affected by age group (p = 0.031 for Aβ42, Supplementary Table 3; p = 0.027 for Aβ40, Supplementary Table 4). Notably, there were no other significant interactions between race and any of the other covariates in models of plasma Aβ levels, suggesting that Black participants had higher mean levels of plasma Aβ42/40 than white participants regardless of amyloid status, sex, APOE ε4 carrier status, years of education, BMI, hypertension or diabetes.

Using the same covariate-adjusted models, levels of the amyloid burden by PET and CSF Aβ42/40 were compared for Black and white groups (Table 2). Notably, the power to discern differences was lower because many fewer individuals underwent amyloid PET or CSF col-lection. The covariate-adjusted mean amyloid PET Centiloid values were slightly lower in 89 Black individuals compared to 626 white individuals, although this did not reach significance (21.6 ± 13.6 Cen-tiloids versus 28.2 ± 13.4 Centiloids, p = 0.072). The covariate-adjusted mean CSF Aβ42/40 values were higher, consistent with lower amyloid burden, in 80 Black individuals compared to 806 white individuals (0.1122 ± 0.0068 versus 0.1069 ± 0.0068, p = 0.014). CSF Aβ42 levels were slightly (8.2%) lower in Black individuals compared to white individuals (840 ± 117 versus 915 ± 116 pg/mL, p = 0.041), but CSF Aβ40 levels were much (23.8%) lower in Black individuals than white indivi-duals (5552 ± 1277 versus 7284 ± 1272 pg/mL, p < 0.0001). CSF t-tau and

**Table 1 | Characteristics of cohort at baseline**

| Characteristics | Overall | | | Cognitively unimpaired | | | Cognitively impaired | | |
|---|---|---|---|---|---|---|---|---|---|
| | Black N = 324[a] | White N = 1547 | p | Black N = 234 | White N = 1027 | p | Black N = 89 | White N = 520 | p |
| **Sample characteristics** | | | | | | | | | |
| Site (n for WU/UPenn/UAB) | 249/63/12 | 1412/115/20 | <0.0001 | 181/47/6 | 934/83/10 | <0.0001 | 67/16/6 | 478/32/10 | <0.0001 |
| Number of plasma samples per individual (n with 1/2/3/ ≥ 4 samples) | 166/74/ 53/31 | 788/340/ 207/212 | 0.15 | 98/58/ 50/28 | 392/258/ 181/196 | 0.056 | 68/16/3/2 | 396/82/ 26/16 | 0.84 |
| Years between the 1st and last plasma sample (mean, SD)[b] | 5.11 (3.52) | 6.93 (4.17) | <0.0001 | 5.22 (3.54) | 7.17 (4.22) | <0.0001 | 4.01 (2.94) | 5.70 (3.67) | 0.018 |
| Fasting status (n for fasting/non-fasting, % fasting) | 117/ 207 (36.1%) | 1015/ 532 (65.6%) | <0.0001 | 91/ 143 (38.9%) | 734/ 293 (71.5%) | <0.0001 | 25/ 64 (28.1%) | 281/ 239 (54.0%) | <0.0001 |
| **Demographics** | | | | | | | | | |
| Baseline age (mean, SD) | 70.2 (8.6) | 70.5 (9.5) | 0.26 | 69.0 (7.78) | 68.3 (9.67) | 0.42 | 73.8 (9.41) | 74.8 (7.55) | 0.40 |
| Sex (n, % female) | 234 (72.2%) | 827 (53.5%) | <0.0001 | 172 (73.5%) | 595 (57.9%) | <0.0001 | 61 (68.5%) | 232 (44.6%) | <0.0001 |
| Years of education (mean, SD) | 15.3 (2.92) | 15.8 (2.81) | 0.002 | 15.4 (2.84) | 16.3 (2.55) | 0.0001 | 14.8 (3.07) | 14.9 (3.06) | 0.71 |
| APOE ε4 status (n for carrier/non-carrier/ missing, % carrier) | 146/172/ 6 (45.1%) | 659/878/ 10 (42.6%) | 0.35 | 96/136/ 2 (41.0%) | 359/667/ 1 (35.0%) | 0.08 | 50/35/ 4 (56.2%) | 300/211/ 9 (57.7%) | 1.00 |
| **Medical conditions** | | | | | | | | | |
| BMI status (n for overweight or obese/ underweight or normal/missing, % overweight or obese) | 225/52/ 47 (69.4%) | 805/426/ 316 (52.0%) | <0.0001 | 179/30/ 25 (76.5%) | 569/282/ 176 (55.4%) | <0.0001 | 46/22/ 21 (51.7%) | 236/144/ 140 (45.4%) | 0.46 |
| Hypertension status (n for positive/ negative/missing, % positive) | 212/108/ 4 (65.4%) | 655/878/ 14 (42.3%) | <0.0001 | 154/79/ 1 (65.8%) | 402/618/ 7 (39.1%) | <0.0001 | 58/29/ 2 (65.2%) | 253/260/ 7 (48.7%) | 0.004 |
| Diabetes status (n for positive/negative/ missing, % positive) | 56/223/ 45 (17.3%) | 93/1150/ 304 (6.0%) | <0.0001 | 44/167/ 23 (18.8%) | 53/805/ 169 (5.2%) | <0.0001 | 12/56/ 21 (13.5%) | 40/345/ 135 (7.7%) | 0.13 |
| Stroke status (n for positive/negative/ missing, % positive) | 6/258/ 60 (1.9%) | 25/1401/ 121 (1.6%) | 0.74 | 3/188/ 43 (1.3%) | 14/945/ 68 (1.4%) | 1.00 | 3/70/ 16 (3.4%) | 11/456/ 53 (2.1%) | 0.63 |

The significance of differences between the Black and white groups are shown. p values were unadjusted from two-sided Wilcoxon rank sum test and Chi-square test for quantitative and categorical characteristics, respectively.

[a]Data on baseline cognitive status is missing for one of N = 324 Black participants.

[b]A sub-cohort had plasma Aβ measures from at least two samples.

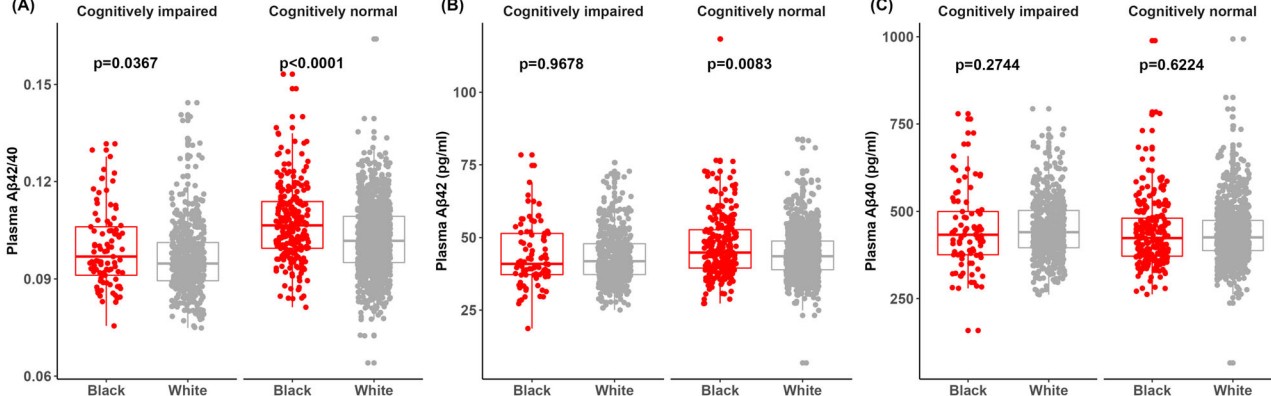

**Fig. 1 | Plasma Aβ values stratified by race and cognitive status.** Values for plasma Aβ42/40 (**A**), Aβ42 (**B**), and Aβ40 (**C**) are shown for Black (N = 323) and white (N = 1547) participants stratified by cognitive status. The box indicates 50% of the data from the 25% quantile to the 75% quantile, with the horizontal lines representing the median. The individual data points were jittered horizontally for better visualization. The whiskers extend to 1.5 times of the 25% and 75% quantile. Overlaid are individual data points with gray for white participants and red for Black participants. The significance of unadjusted racial differences was evaluated by two-sided Wilcoxon rank sum tests with unadjusted p values reported. Source data are provided as a Source Data file.

p-tau181 levels were less abnormal in Black individuals compared to white individuals (t-tau: 114.2 ± 83.1 versus 198.1 ± 82.9 pg/mL, p = 0.001; p-tau181: 17.42 ± 12.31 versus 28.46 ± 12.27 pg/mL, p = 0.004).

### Racial differences in the relationships of plasma biomarkers with CSF biomarkers, amyloid PET, and cognitive composite scores

Spearman correlations of plasma Aβ42/40, Aβ42 and Aβ40 with CSF biomarkers, amyloid PET, and cognitive composite scores were

examined within each group and compared across Black and white groups (Fig. 2). For both Black and white groups, the strongest correlation was between plasma Aβ42/40 and CSF Aβ42/40, but Black individuals had a weaker correlation (ρ = 0.51 versus 0.64, difference −0.13, raw p < 0.05). This racial difference persisted after adjustment for covariates (ρ = 0.38 versus 0.49, difference −0.11, raw p < 0.05) (Supplementary Fig. 1), but was no longer significant after FDR-based multiplicity adjustment. In fact, no significant differences in the partial correlations of plasma Aβ biomarkers with CSF biomarkers and

**Table 2 | Adjusted mean baseline biomarker values for groups of Black and white individuals**

| Biomarker | Group | N | Mean adjusted value | Standard error | p |
|---|---|---|---|---|---|
| Plasma Aβ42/Aβ40 | Black | 214 | 0.1201 | 0.0030 | |
| | White | 1113 | 0.1155 | 0.0030 | |
| | Difference (Black–white) | | 0.0046 | 0.0008 | <0.0001 |
| Plasma Aβ42 (pg/mL) | Black | 214 | 26.25 | 2.40 | |
| | White | 1113 | 26.41 | 2.36 | |
| | Difference (Black–white) | | −0.16 | 0.61 | 0.80 |
| Plasma Aβ40 (pg/mL) | Black | 214 | 177.8 | 22.7 | |
| | White | 1113 | 199.5 | 22.4 | |
| | Difference (Black–white) | | −21.6 | 5.7 | 0.0002 |
| Amyloid PET Centiloid | Black | 89 | 21.6 | 13.6 | |
| | White | 626 | 28.2 | 13.4 | |
| | Difference (Black–white) | | −6.6 | 3.7 | 0.072 |
| CSF Aβ42/40 | Black | 80 | 0.1122 | 0.0068 | |
| | White | 806 | 0.1069 | 0.0068 | |
| | Difference (Black–white) | | 0.0053 | 0.0021 | 0.014 |
| CSF Aβ42 (pg/mL) | Black | 80 | 840 | 117 | |
| | White | 806 | 915 | 116 | |
| | Difference (Black–white) | | −75 | 37 | 0.041 |
| CSF Aβ40 (pg/mL) | Black | 80 | 5552 | 1277 | |
| | White | 806 | 7284 | 1272 | |
| | Difference (Black–white) | | −1732 | 400 | <0.0001 |
| CSF t-tau (pg/mL) | Black | 80 | 114.2 | 83.1 | |
| | White | 806 | 198.1 | 82.9 | |
| | Difference (Black–white) | | −84.0 | 26.1 | 0.001 |
| CSF p-tau181 (pg/mL) | Black | 80 | 17.42 | 12.31 | |
| | White | 806 | 28.46 | 12.27 | |
| | Difference (Black–white) | | −11.04 | 3.86 | 0.004 |

The linear regression models included the main effects of race and the covariates of age, sex, *APOE* ε4 carrier status, years of education, cognitive status (unimpaired, CDR 0; or impaired, CDR > 0), fasting status if plasma biomarker (fasting or non-fasting), BMI, and status for hypertension, diabetes, and stroke (positive or negative). The two-sided *t*-test raw *p*-values testing for the significance of differences between the Black and white groups are shown.

amyloid PET and cognitive measures were found across racial groups after the multiplicity adjustment.

### Racial differences in the longitudinal changes of plasma Aβ biomarkers

A sub-cohort of 158 Black participants and 759 white participants had plasma Aβ measures from at least two samples with a mean interval between the first and last plasma sample of 5.11 ± 3.52 years for Black individuals and 6.93 ± 4.17 years for white individuals. Longitudinal trajectories of plasma Aβ42, Aβ40, and Aβ42/40 appeared relatively linear (Fig. 3), justifying the use of linear models (Supplementary Fig. 2). Covariate-adjusted linear mixed-effects models were used to estimate the rate of change of plasma Aβ in 112 Black and 566 white individuals (Table 3). Plasma Aβ42/40 decreased longitudinally in Black and white individuals at a rate that did not vary by race (p = 0.38).

However, plasma Aβ42 increased longitudinally at a faster rate in Black compared to white individuals (0.5719 ± 0.1142 versus 0.2642 ± 0.0495 pg/mL/year, p = 0.013). Plasma Aβ40 also increased at a faster rate in Black compared to white individuals (7.357 ± 1.047 versus 5.464 ± 0.454 pg/mL/year, p = 0.093), although this difference did not reach statistical significance.

Further analyses examined whether racial differences in the rate of change of plasma Aβ were modified by baseline amyloid status, age, sex, *APOE* ε4 carrier status, years of education, cognitive status, BMI, hypertension, or diabetes. Amyloid status by CSF or amyloid PET did not affect racial differences in the rate of change of plasma Aβ42/40, Aβ42, or Aβ40 (Supplementary Table 5). Further, no other covariates significantly affected racial differences in the rate of change of plasma Aβ42/40 (Supplementary Table 6). However, for plasma Aβ42 and Aβ40, there was a significant interaction between the racial group and baseline age such that younger but not older Black participants had a faster increase in Aβ42 (p = 0.0059, Supplementary Table 7) and Aβ40 (p = 0.018, Supplementary Table 8) than the white counterparts. Overall, plasma Aβ42/40 changed relatively consistently across racial groups despite multiple potentially confounding factors.

## Discussion

This study examined potential differences in baseline levels and rates of longitudinal change in plasma Aβ measures (Aβ42, Aβ40, and Aβ42/40) in self-identified Black and white participants, in one of the largest cohorts studied so far. Black participants had higher average plasma Aβ42/40 levels at baseline than white participants, consistent with Black participants having a lower average level of amyloid pathology. Despite the baseline differences, the Black and white individuals had similar longitudinal rates of change in plasma Aβ42/40, consistent with a similar rate of amyloid accumulation in both groups.

The finding of the lower average level of amyloid pathology in Black individuals compared to white individuals aligns with three recent CSF and imaging biomarker studies. One imaging study of 144 Black and 3689 white cognitively normal individuals reported that the Black participants had a lower rate of amyloid positivity and lower average amyloid burden[15]. A second imaging study with 635 Black and 15,322 white cognitively impaired individuals reported that Black participants were less likely to be amyloid PET positive[17]. In a cohort overlapping with the current study cohort that included 266 Black and 1977 white participants with CSF biomarkers, Black participants had less abnormality of multiple CSF biomarkers, including Aβ42/40, total tau, p-tau181, and neurofilament light[21]. Other studies have found the opposite result or no differences between these groups[22,23]. It is possible that differences in study findings could be related to variable methods of recruitment[37]. Still, growing evidence suggests that Black individuals have a lower average level of amyloid pathology compared to white individuals, at least within the population of individuals in AD research studies and clinical trials.

In our current study, Black participants had higher mean levels of plasma Aβ42/40 than white participants regardless of amyloid status, sex, *APOE* ε4 carrier status, years of education, BMI, hypertension, or diabetes status. However, the possibility remains that variables not currently in our dataset, including additional medical conditions (e.g., chronic kidney disease) and/or social and structural determinants of health, might explain these racial differences[38]. Until these factors are identified and included in datasets for analyses, the significant effect of race in models of plasma Aβ42/40, despite adjustment for key covariates, implies that statistical analyses of plasma AD biomarkers are more valid when race is included as a variable. Moreover, the converging findings across plasma biomarkers, CSF biomarkers, and amyloid PET suggest that Black individuals may truly have a lower average level of amyloid pathology compared to white individuals for reasons that we do not yet fully understand.

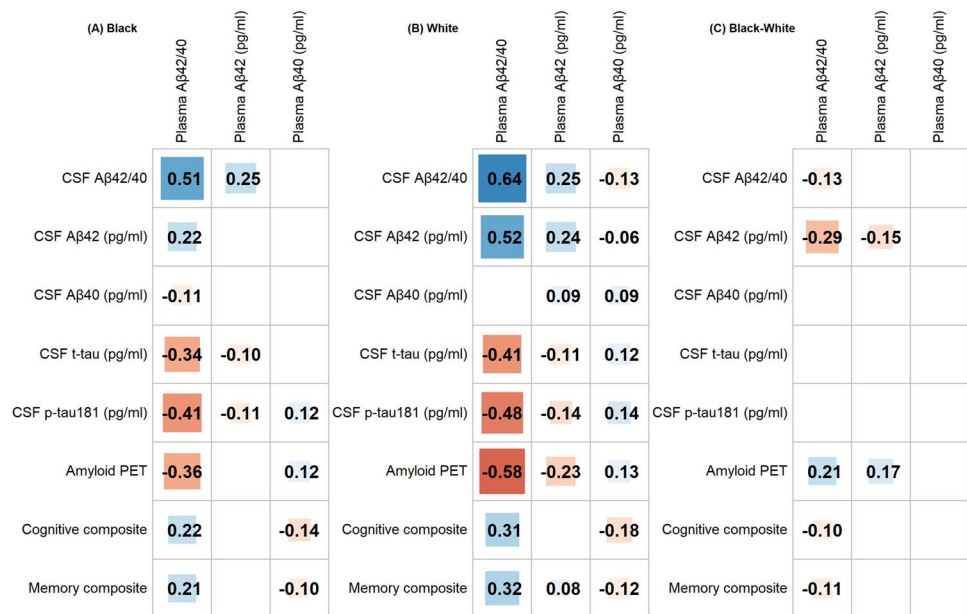

**Fig. 2 | Correlations of plasma Aβ biomarkers with CSF biomarkers, amyloid PET, and cognitive composites.** Spearman correlations between plasma Aβ biomarkers and CSF biomarkers, amyloid PET, or cognitive composite are shown for Black (**A**) and white (**B**) individuals. Racial differences in correlations (Black–white) are shown in (**C**). Only significant correlations or differences (raw unadjusted $p < 0.05$) from the two-sided correlation test are shown. Colored blocks visualize correlation direction and magnitude, with warm to cold colors representing most negative correlations to most positive correlations and darker colors and larger block sizes corresponding to greater absolute magnitudes. Source data are provided as a Source Data file.

**Table 3 | Estimated annual rates of longitudinal change in plasma Aβ**

| Biomarker | Black N = 112 | | | White N = 566 | | | Difference (Black–white) | | |
|---|---|---|---|---|---|---|---|---|---|
| | Slope | SE | p | Slope | SE | p | Slope | SE | p |
| Plasma Aβ42/40 | −0.00047 | 0.0001 | 0.002 | −0.00060 | 0.00006 | <0.0001 | 0.00013 | 0.00015 | 0.38 |
| Plasma Aβ42 | 0.5719 | 0.1142 | <0.0001 | 0.2642 | 0.0495 | <0.0001 | 0.3076 | 0.1226 | 0.013 |
| Plasma Aβ40 | 7.357 | 1.047 | <0.0001 | 5.464 | 0.454 | <0.0001 | 1.893 | 1.125 | 0.093 |

The linear mixed-effects models included the main effects of race and time, and race by time interaction, as well as the covariates of age, sex, *APOE ε4* carrier status, years of education, cognitive status (unimpaired, CDR 0; or impaired, CDR > 0), fasting status (fasting or non-fasting), BMI, and status for hypertension and diabetes (positive or negative). Whether the slope is significantly different from zero for Black and white participants, and the significance of racial differences from two-sided *t*-test are shown.

Lower average levels of amyloid pathology in Black individuals may have a major impact on efforts to make AD clinical trials more representative. A recent study of four AD clinical trials that required amyloid positivity for inclusion found much lower eligibility of Black individuals[18]. There have been suggestions to decrease the level of amyloid biomarker abnormality required for Black individuals to increase enrollment in this under-represented group. However, the use of race-specific cut-offs could have unintended consequences and even lead to systematic racial discrimination[39]. For example, if Black individuals included in clinical trials had less amyloid pathology, they may be less responsive to amyloid-lowering treatments, potentially leading to the erroneous conclusion that the treatments are not effective in Black individuals. These concerns provide a strong rationale for not using race-specific cut-offs, even if the intent is to increase the representation of Black individuals in AD clinical trials.

Interestingly, the higher average plasma Aβ42/40 levels in Black participants resulted from lower levels of plasma Aβ40 but similar levels of plasma Aβ42 compared to white participants. Aligned with this finding, CSF Aβ40 levels were much lower in Black participants compared to white participants. Importantly, it is the ratio of Aβ42 to Aβ40 that reflects sequestration of Aβ42 into amyloid plaques and is most strongly associated with amyloid plaque burden[40,41]. Recent work has found that individuals have significant differences in overall levels

of CSF proteins that are driven by non-AD related factors and that CSF Aβ40 is a useful measure of an individual's overall CSF protein level[42,43]. The reasons for differences in overall levels of CSF proteins are not yet well understood but may be related to physiological factors in CSF production and clearance that are associated with age, sex, ventricular volumes, and circadian rhythms[43,44]. Additional factors may influence the proportion of central nervous system-derived proteins that reach the plasma[45]. Normalizing fluid biomarkers of AD (e.g., CSF p-tau181) to Aβ40 may decrease non-AD-related variance and improve associations with AD pathology[42,43]. Our findings suggest that Black individuals have lower levels of CSF and plasma Aβ40. It is possible that normalization to Aβ40 may reduce racial differences in biomarkers of AD pathology[16,28].

An important finding of this study was that the longitudinal rate of change in plasma Aβ42/40 did not vary significantly between groups of Black and white individuals, despite racial differences in baseline plasma Aβ42/40. While this finding must be replicated in other cohorts, this suggests that while a lower average level of amyloid pathology may result in lower enrollment of Black participants in studies and trials that use biomarkers of amyloid pathology as inclusion criteria, once participants are enrolled and randomized, changes in plasma Aβ42/40 may be consistent across racial groups. Furthermore, changes in plasma Aβ42/40 were not differentially affected by

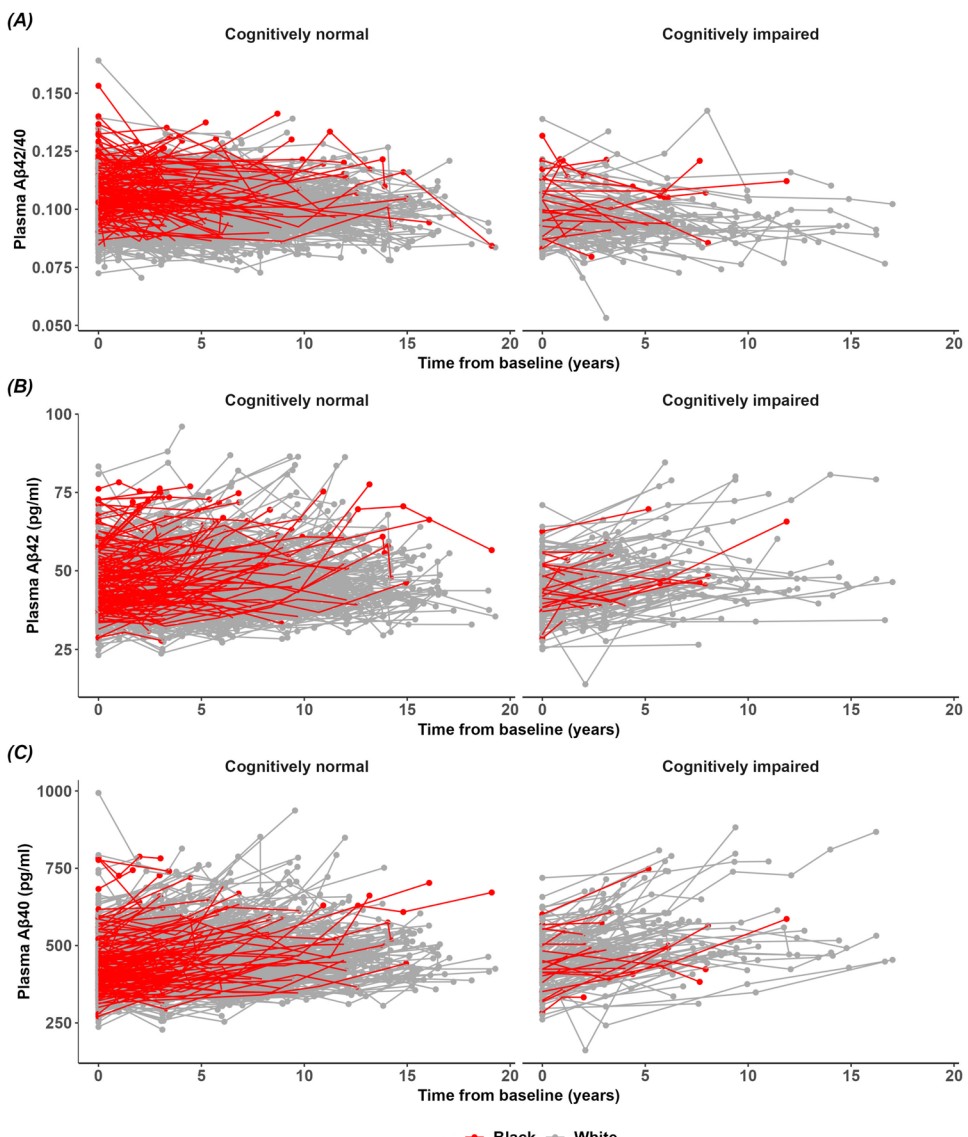

**Fig. 3 | Spaghetti plots of plasma biomarkers stratified by cognitive status.**
Plasma Aβ42/40 (panel **A**), Aβ42 (panel **B**), and Aβ40 (panel **C**) were plotted against time since baseline between cognitively normal (CDR 0, left panel) and impaired (CDR > 0, right panel) Black ($N = 158$) and white ($N = 759$) participants. Each line represents an independent individual connecting longitudinal data points, gray for white participants and red for Black participants.

race for most covariates except cognitive status. The consistency in the rate of change may allow plasma Aβ42/40 to be used in cohorts with Black and white participants to establish the effect of treatments on biomarkers. Specifically, the placebo arm in future clinical trials may estimate the same rate of change in plasma Aβ42/40 for both Black and white groups.

This study has multiple major strengths. The cohort included a relatively large number of Black participants from three sites that had similar inclusion and exclusion criteria and used a uniform clinical assessment protocol[46]. Plasma samples were collected according to standard protocols that did not vary by racial group, and samples were assayed together at a single lab. The plasma Aβ assay, PrecivityAD™, was previously shown to accurately and consistently classify amyloid status in an overlapping cohort with Black and white participants[16]. This test is currently being used in clinical trials as well as in clinical care[36], making our results of interest to researchers, clinical trialists, and clinicians. Limitations of our study include that AD research cohorts are not representative of the general population and do not represent a random sample from the community. There was very limited data on structural and social

determinants of health, including socioeconomic status, especially life course experience, and discrimination. Some individuals were missing data on covariates, reducing the number of individuals in covariate-adjusted analyses and, hence, statistical power. Further, data on more recently developed plasma p-tau217 assays were not currently available for most samples in our cohort; an examination of racial differences in plasma p-tau217 is planned once sufficient data is available. Finally, it is possible that an even larger cohort with longer longitudinal follow-up might reveal racial differences in the rate of change of plasma Aβ42/40.

In summary, we found that Black research participants have higher average plasma Aβ42/40 at baseline, consistent with less amyloid pathology, compared to white participants. Interestingly, despite these racial differences at baseline, the rate of change of plasma Aβ42/40 was consistent in both Black and white groups. Further, plasma Aβ42/40 had relatively consistent associations with CSF Aβ42/40, amyloid PET, and cognitive measures across both groups. These results suggest that plasma Aβ42/40 may be useful in providing a biomarker outcome for research and clinical trials that is consistent across racial groups.

## Methods

### Ethical approval

All participants provided written informed consent at recruitment from their parent studies. The Washington University Human Research Protection Office approved the current study with additional approvals from the Institutional Review Boards of the other sites.

### Participants

The study cohort included individuals with plasma Aβ measures and clinical/cognitive data who participated in the Study of Race to Understand Alzheimer Biomarkers (SORTOUT-AB; NIH/NIA R01 AG067505), which aims to understand potential racial differences in harmonized biomarker data collected by multiple research studies of memory and aging in middle-aged and older individuals[21]. Participants in the current study represented three of the SORTOUT-AB sites: the Washington University (WU) Knight Alzheimer's Disease Research Center (ADRC), the University of Pennsylvania (UPenn) ADRC, and the University of Alabama at Birmingham (UAB) ADRC. Details of recruitment for these studies have been described previously[20,21,47]. Participants with conditions that could prevent participation or affect long-term participation (e.g., metastatic cancer) were excluded. Participants underwent clinical and/or cognitive assessments within 2 years of their baseline plasma assessments. A sub-cohort of the participants also had CSF or imaging assessments within 2 years of their baseline plasma sample collection.

### Clinical and cognitive assessments

Clinical and cognitive assessments from WU, UPenn, and UAB followed protocols consistent with the National Alzheimer's Coordinating Center Uniform Data Set (UDS)[46]. Demographic information, body mass index (BMI), and medical history were collected. Race and sex were self-identified by participants. Cognitive impairment was scored with the Clinical Dementia Rating®™ (CDR®™)[48]. Individuals with a CDR of 0 were categorized as cognitively unimpaired. Individuals with a CDR of 0.5 or greater were categorized as cognitively impaired, and the probable etiology was formulated by clinicians based on clinical features in accordance with standard criteria[49]. The cognitive battery of the UDS included tasks of episodic memory, working memory, semantic knowledge, executive function and attention, and visuospatial ability, and were harmonized across UDS versions[50]. Global cognitive and episodic memory composite scores were calculated as previously described[21].

### Apolipoprotein E genotyping

Apolipoprotein E (APOE) genotyping was performed as previously described[51]. Participants were classified as APOE ε4 carriers (one or two ε4 alleles) and non-carriers.

### Blood collection and analysis

At WU, blood was collected from non-fasting participants at the time of clinical assessment or fasting participants at the time of lumbar puncture (LP)[52]. Blood was collected from non-fasting participants at UPenn and fasting participants at UAB at the time of clinical assessment. Blood was collected in EDTA-containing tubes and centrifuged to separate plasma from blood cells. Plasma was aliquoted into polypropylene tubes and stored at −80 °C until analysis.

All plasma samples were analyzed at $C_2N$ Diagnostics with the PrecivityAD assay, which has previously been described[53,54]. Briefly, Aβ40 and Aβ42 were simultaneously immunoprecipitated from plasma via a monoclonal anti-Aβ mid-domain antibody[53]. Proteins were digested into peptides using LysN endoprotease. Liquid chromatography–mass spectrometry was performed on a Thermo Scientific Orbitrap Lumos Tribrid mass spectrometer interfaced with a nano-Acquity chromatography system (LC–MS/MS)[53].

### CSF collection and analysis

At WU, CSF samples (20–30 mL) were collected at 8 AM after overnight fasting by gravity drip, briefly centrifuged at low speed, and aliquoted into polypropylene tubes prior to freezing at −80 °C. CSF samples from participants enrolled at the UPenn and UAB ADRCs were collected in accordance with protocols for the Alzheimer's Disease Neuroimaging Initiative (ADNI)[55].

An automated immunoassay (LUMIPULSE G1200, Fujirebio, Malverne, PA) was used to measure CSF concentrations of Aβ40, Aβ42, total tau (t-tau), and tau phosphorylated at position 181 (p-tau181)[56,57]. A bridging subset of the CSF samples ($n = 114$) from the UPenn ADRC was selected to represent a wide range of values for all analytes and was run at the same time and with the same reagents as the WU samples to evaluate and adjust for systematic differences between the WU and UPenn sites. A model fitted on the values of the bridging samples was used to harmonize the CSF biomarker values between WU and UPenn[58]. Positive CSF biomarker status was defined as a CSF Aβ42/40 value < 0.0673[57].

### Imaging processing and analysis

Structural brain MRI and amyloid PET protocols were consistent with those used by the ADNI[21]. A standardized uptake value ratio (SUVR) with correction for partial volume effects was calculated for the FreeSurfer regions of interest (ROIs) for PiB, Florbetapir, or Florbetaben[59]. The cerebellum was used as the reference region. A summary measure of amyloid burden was calculated using the averaged SUVR values in the lateral orbitofrontal, medial orbitofrontal, precuneus, rostral middle frontal, superior frontal, superior temporal, and middle temporal regions. To harmonize SUVR values across different tracers (PiB, Florbetapir, or Florbetaben), values from the summary measure were converted into Centiloid units[60]. Positive amyloid PET status was defined as a Centiloid value > 20[36].

### Statistical analyses

The baseline characteristics of participants were summarized with the mean and SD for continuous variables or count and percentage for categorical variables. General linear models were implemented to evaluate for cross-sectional racial differences in levels of fluid or imaging biomarkers. These models included the main effects of race and the covariates of age, sex, APOE ε4 carrier status, years of education, cognitive status (unimpaired, CDR 0; or impaired, CDR > 0), fasting status if plasma biomarker (fasting or non-fasting), BMI, and status for hypertension, diabetes, and stroke (positive or negative). Additional analyses included interactions of each variable with the racial group to examine whether each of the covariates modified the racial differences; stroke status was not included in these interaction models due to a small number of individuals with stroke.

Fluid biomarker measures were correlated with the established AD biomarkers by Spearman correlations, and the correlations between groups were compared by a two-sided standard normal test after Fisher's Z-transformation[61]. Because of the large number of comparisons in the correlations, we adjusted statistical significance for a false discovery rate (FDR)[62] of 5%.

General linear mixed models with random slopes and intercepts were implemented to evaluate for racial differences in the longitudinal rates of change of fluid biomarker measures[63]. All models included race and a race-by-time interaction (at baseline time = 0), and the same covariates as in the cross-sectional models. The annual rates of change between groups were compared by a two-sided approximate Student t-test; the degree of freedom was estimated by the Satterthwaite method. All models were implemented in Rstudio (version 2023.9.1.494 running R version 4.2.1) via the R package lmerTest (version 3.1-3). An assessment found no clear non-linear longitudinal patterns, likely because of the small number of plasma samples for most individuals.

## Reporting summary

Further information on research design is available in the Nature Portfolio Reporting Summary linked to this article.

## Data availability

Anonymized data that support the findings of this study are available from the corresponding author and co-first authors upon request from qualified investigators. Briefly, a formal application should be submitted to provide the rationale and objectives of the research. Data will be provided after an application is reviewed and approved. The study participants provided their data with these conditions for use of the data. Source data are provided with this paper.

## Code availability

Publicly available software, Rstudio, was used for all analyses as described in the "Methods" section and the accompanying Reporting Summary.

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

## Acknowledgements

The authors thank the research volunteers who participated in the studies from which these data were obtained and their families. The authors express gratitude to Dr. Anne M. Fagan for establishing the large WU biorepository used in this study. We acknowledge the WU, UPenn, and UAB ADRC biospecimen cores. The authors thank C$_2$N Diagnostics for processing the plasma samples and conducting the QC of the plasma biomarker data. This work was supported in part by funding from the National Institutes of Health grants R01 AG067505 (PI: Xiong), R01 AG070941 (PI: Schindler), P30 AG066444 (PI: Holtzman), P01 AG026276 (PI: Morris), P01 AG003991 (PI: Morris), P30 AG072979 (PI: David Wolk), P20 AG068024 (PI: Erik Roberson), and R44 AG059489 (C$_2$N Diagnostics). Additional funding was provided by BrightFocus (CA2016636), the Gerald and Henrietta Rauenhorst Foundation, the Cure Alzheimer's Fund (PI: Moulder), and the Alzheimer's Drug Discovery Foundation (GC-201711-2013978).

## Author contributions

C.X., J.L., D.A.W., L.M.S., E.D.R., T.L.S.B., R.J.B., C.C., J.C.M., and S.E.S. contributed to the conception and design of the study. C.X., D.A.W., L.M.S., E.D.R., C.F.M., R.L.H., T.L.S.B., Q.B., F.A., E.G., E.N.G., K.L.M., D.S.G., O.J.C., C.C., D.M.H., R.J.B., J.C.M., and S.E.S. contributed to the acquisition and analysis of data. C.X., J.L., C.H.M., R.L.H., Q.B., F.A., E.N.G., K.L.M., D.S.G., O.J.C., G.B., C.C., D.M.H., R.J.B., J.C.M., and S.E.S. contributed to drafting the text or preparing figures.

## Competing interests

D.A.W. has served as a paid consultant for Eli Lilly, GE Healthcare and Qynapse, and serves on a DSMB for Functional Neuromodulation. E.D.R. serves on a data monitoring committee for Eli Lilly. T.L.S.B. participates as a site investigator in clinical trials sponsored by Avid Radiopharmaceuticals, Eli Lilly and Company, Biogen, Eisai, Janssen, and Roche. D.S.G. participates as a site investigator in clinical trials sponsored by Biogen and Janssen. He serves as a consultant to Eisai, Lilly, and Roche. DMH and RJB co-founded and have equity in C$_2$N Diagnostics. D.M.H. serves on the scientific advisory board of C2N Diagnostics, Genentech, Denali, Cajal Neurosciences, and Asteroid. S.E.S. has served on advisory boards for

Eisai. Washington University has a financial interest in $C_2N$ Diagnostics and may financially benefit if the company is successful in marketing its product(s) that is/are related to this research. The other authors declare no competing interests.
