## [Peer Review File · Nature Communications]

Baseline levels and longitudinal changes in plasma A β 42/40 among Black and white individualsREVIEWER COMMENTS

Reviewer #1 (Remarks to the Author):

This is an interesting and relevant manuscript evaluating the performance of the new AD plasma related markers (i.e., Aβ42/40 ratio) in participants that are often underrepresented in research studies. The topic is highly important; however the manuscript would be highly improved by inclusion of additional analysis, more specifically:

1. Specify the characteristics of the groups of analysed (i.e., cognitively unimpaired, dementia cases?). It seems that all cases are analysed together. Since staging may affect biomarker performance, sensitivity analysis stratified per stage group should be included. Staging may also influence important measures of this study, such as longitudinal changes on plasma biomarker levels. Such analysis should be also included.
2. Were not CSF or PET data in a subset of cases available to also confirm the performance of the markers to detect amyloid pathology? This should be present for at least a subset of cases. The lack of such data limits the interpretation that Black individuals have lower amyloid pathology considering that, as these were understood, the gold reference standard was established using white individuals. Such analysis is highly important especially considering the discordant results observed in previous studies, as also highlighted by the authors. This may thus impact the subsequent conclusion that fewer proportion of Black individuals might be eligible for clinical trials.
3. Other plasma markers that are highly relevant for AD diagnosis and will be likely implemented in clinical routine (e.g., pTau217 or 181). The relevance of the manuscript will be highly increased by the analysis of these markers, as the biomarker analysed is mostly being used within US settings only.
4. For visual purposes, plots with biomarkers levels would help. Same applies for correlations, which could be added as supplementary.
5. The lack of significant changes in the rate of change is not surprising, considering the short time span and that groups with different cognitive deficits are analysed together.

Reviewer #2 (Remarks to the Author):

In this work, Xiong and colleagues investigated differences in baseline and longitudinal levels of plasma Aβ42, Aβ40 and Aβ42/40 between African American and non-Hispanic White individuals using linear (mixed) models, and examined the influence of multiple covariates. They showed that plasma Aβ40 but not Aβ42 levels were lower at baseline in African American individuals compared to White participants, resulting in a higher Aβ42/40 as well. Longitudinal changes in Aβ40 and Aβ42/40 were not different between racial groups, while Aβ42 increases over time were larger in African American participants.

The authors have written the paper in a clear way and the analyses are sound. This study is one of the first in its kind, with unique data. Its findings are highly relevant for research, trial and clinical settings and as such I believe it will make an important contribution to the field.

There are some concerns and questions that I would like to address:

1. Additional cohort details:

a. While the manuscript provides information about the time interval between first and last plasma sample collected, it is unclear how many samples were collected on average (mean (SD) or range) and in case of >2 samples what the average time between each sample collection was.

b. In the methods, the authors describe that clinical and cognitive assessments were performed within two years of participants' baseline plasma collection. Two years seems quite a long time during which many factors may change (cognition, comorbidities, etc.). While I can imagine this interval was often much shorter, a description of the average time interval between these two (and possible differences between the racial groups) would help with interpretation of the findings.

2. One of the main conclusions is that higher plasma A β 42/40 in African American participants was consistent with less amyloid pathology. However, no comparison was made to for example amyloid load on PET, no information on number (%) of amyloid-positive or amyloid-negative individuals, and no autopsy findings were available to support these findings. Furthermore, would it be possible that non-AD related factors are the cause of lower A β 40 levels, and as a result higher A β 42/40 as well? Would perhaps racial-specific cut-points be necessary to determine amyloid pathology? My advice would be to nuance the conclusion and add to the discussion on possible alternative explanations for these findings.

3. While the main focus of the paper is longitudinal plasma biomarkers, this topic was slightly underrepresented in the results section. The first paragraph of the results (cohort characteristics) could be shortened, while paragraph 3.4 could be expanded:

a. I was surprised to see that A β 42 increased in Black (and White?) participants, while I would expect these levels to decrease. Do you have an explanation for these findings? And does A β 42/40 increase or decrease in each group?

b. It would help interpret these findings if you could include a table with the results of the linear mixed models (including β (SE) estimates for biomarker slopes for both Black and White participants), and perhaps a figure as well.

c. In CSF, A β levels are known to plateau once amyloid becomes abnormal, and the rate of change differs depending on amyloid status. I would be interested in seeing a subgroup-analysis of longitudinal changes in plasma A β 42, A β 40 and A β 42/40 levels in A- vs A+ individuals (based on CSF or PET) in both African American and White participants.

d. Furthermore, for those who have CSF and PET available: how many participants were A- and A+?

4. Last, the influence of most covariates on baseline and longitudinal A β levels in both races was examined, except for the influence of hypertension and diabetes mellitus. Why did the authors choose to exclude those factors from these analyses? I would be interested in seeing the results, especially as hypertension and diabetes are a risk factor for chronic kidney disease, which is known to influence plasma biomarker levels. Also: where measures of kidney functioning avail

Reviewer #3 (Remarks to the Author):

This study examined a large biracial AD research cohort to evaluate for potential differences in baseline levels and rates of longitudinal change in plasma A β measures (A β 42, A β 40, and A β 42/40) in self-identified Black and White participants. It was found that Black participants had a higher average baseline levels of plasma A β 42/40 than White participants, which was due to lower average baseline levels of plasma A β 40 (166.10 vs. 187.72 pg/mL; p=0.0004). Plasma A β 42/40 was significantly correlated with almost all CSF and amyloid PET biomarkers as well as cognitive scores in White participants, and the correlations were largely consistent between Black and White participants. There were no significant racial differences in the rate of change in A β 42/40 and A β 40, but the Black group had a faster rate of increase in A β 42 compared to the White group. These findings may suggest that a lower proportion of Black subjects might be eligible for research studies/clinical trials; however, the rate of change of plasma A β 42/40 was consistent across racial groups. These are interesting and significant findings. The paper could be improved by some changes:

- 1) The authors might speculate as to why the Black participants might have a lower baseline A β 40 level.
- 2) There should be clarification that Black and White participants across the sites represent a random sample from the community. Selection factors and/or inclusion/exclusion criteria must be similar across centers and no temporal differences exist between the ascertainment of measures between Whites and Blacks, to ensure that these are not confounding factors.

Reviewer #1 (Remarks to the Author):

The topic is highly important; however the manuscript would be highly improved by inclusion of additional analysis, more specifically:

1. Specify the characteristics of the groups of analysed (i.e., cognitively unimpaired, dementia cases?). It seems that all cases are analysed together. Since staging may affect biomarker performance, sensitivity analysis stratified per stage group should be included. Staging may also influence important measures of this study, such as longitudinal changes on plasma biomarker levels. Such analysis should be also included.

In our original analyses, we included CDR as a main covariate and carefully examined whether the racial differences varied by CDR by testing the interaction between race and CDR on both intercept (baseline level at time=0) and slope (annual rate of change). We have now revised Table 1 to stratify the demographics and APOE genotypes by cognitive status, e.g., cognitively unimpaired vs. cognitively impaired. Figures of plasma A β values are stratified by cognitive status (Figure 1 and Figure 3). We have also reported the effects of cognitive status on racial differences in newly added Supplementary Tables 2-4, 6-8.

2. Were not CSF or PET data in a subset of cases available to also confirm the performance of the markers to detect amyloid pathology? This should be present for at least a subset of cases. The lack of such data limits the interpretation that Black individuals have lower amyloid pathology considering that, as these rewire understood, the gold reference standard was established using white individuals. Such analysis is highly important especially considering the discordant results observed in previous studies, as also highlighted by the authors. This may thus impact the subsequent conclusion that fewer proportion of Black individuals might be eligible for clinical trials.

Thank you for this suggestion. Although the numbers are lower, we have now added this to Table 2 and our results: "Using the same covariate adjusted models, levels of amyloid burden by PET and CSF A β 42/40 were compared for Black and White groups (Table 2). Notably, the power to discern differences was lower because many fewer individuals underwent amyloid PET or CSF collection. The covariate-adjusted mean amyloid PET Centiloid values were slightly lower in 89 Black individuals compared to 626 White individuals, although this did not reach significance (21.6 ± 13.6 Centiloids versus 28.2 ± 13.4 Centiloids, $p=0.072$). The covariate-adjusted mean CSF A1342/40 values were higher, consistent with lower amyloid burden, in 80 Black individuals compared to 806 White individuals (0.1122 ± 0.0068 versus 0.1069 ± 0.0068 , $p=0.014$). CSF A β 42 levels were slightly (8.2%) lower in Black individuals compared to White individuals (840 ± 117 pg/mL versus 915 ± 116 pg/mL, $p=0.041$), but CSF A β 40 levels were much (23.8%) lower in Black individuals than White individuals ($5,552 \pm 1,277$ pg/mL versus $7,284 \pm 1,272$ pg/mL, $p<0.0001$). CSF t-tau and p-tau181 levels were less abnormal in Black individuals compared to White individuals (t-tau: 114.2 ± 83.1 pg/mL versus 198.1 ± 82.9 pg/mL, $p=0.001$; p-tau181: 17.42 ± 12.31 pg/mL versus 28.46 ± 12.27 pg/mL, $p=0.004$)." (page 12)

3. Other plasma markers that are highly relevant for AD diagnosis and will be likely implemented in clinical routine (e.g., pTau217 or 181). The relevance of the manuscript will be highly increased by the analysis of these markers, as the biomarker analysed is mostly being used within US settings only.

We agree that evaluating plasma p-tau biomarkers is extremely important and we are actively working with C2N Diagnostics on running these assays. However, we currently have only a subset of samples with plasma p-tau values, and very little longitudinal p-tau data from Black individuals. It will take many more months before sufficient data are available from C2N Diagnostics to power these analyses. We think our current findings on plasma A β 42/40 are relevant to clinical trials and clinical practice now, and therefore don't want to wait to publish our findings until the plasma p-tau data is available. We also think the newly added analysis on CSF p-tau181 (page 12 and Table 2) as suggested by the reviewer in the previous comment, which also indicated lower abnormality of Black participants (in comparison to White participants), is not only consistent with our plasma A β 42/40 findings, but also relevant to the reviewer's comment. Finally we have now stated "Further, data on more recently developed plasma p-tau217 assays were not currently available for most samples in our cohort; an examination of racial differences in plasma p-tau217 is planned once sufficient data is available." (page 18)

4. For visual purposes, plots with biomarkers levels would help. Same applies for correlations, which could be added as supplementary.

Thank you for this suggestion. We have added boxplots of baseline plasma A β levels (Figure 1) and two correlation plots (Figure 2 and Supplementary Figure 1).

5. The lack of significant changes in the rate of change is not surprising, considering the short time span and that groups with different cognitive deficits are analysed together.

The longitudinal models did include cognitive status, and we found no modifying effect of cognitive status on racial differences in plasma A β in newly added Supplementary Table 6-8. We do agree, however, that follow-up is a limitation and have added this to our discussion, "Finally, it is possible that an even larger cohort with longer longitudinal follow-up might reveal racial differences in the rate of change of plasma A β 42/40." (page 18)

Reviewer #2 (Remarks to the Author):

There are some concerns and questions that I would like to address:

1. Additional cohort details:

a. While the manuscript provides information about the time interval between first and last plasma sample collected, it is unclear how many samples were collected on average (mean (SD) or range) and in case of >2 samples what the average time between each sample collection was.

Thank you for this suggestion. To Table 1 we have added, "Number of plasma samples per individual (n with 1/2/3/ \geq 4 samples)" and "Follow up years (mean, SD)." We have

also added the time intervals to the longitudinal section of the results, “A sub-cohort of 158 Black participants and 759 White participants had plasma A β measures from at least two samples with a mean interval between the first and last plasma sample of 5.11 ± 3.52 years for Black individuals and 6.93 ± 4.17 years for White individuals.” (page 13).

b. In the methods, the authors describe that clinical and cognitive assessments were performed within two years of participants' baseline plasma collection. Two years seems quite a long time during which many factors may change (cognition, comorbidities, etc.). While I can imagine this interval was often much shorter, a description of the average time interval between these two (and possible differences between the racial groups) would help with interpretation of the findings.

Thank you for this suggestion. The interval was indeed much shorter, and this has been added to the results, “The time interval between clinical assessment and plasma collection was $0.16 \text{ years} \pm 0.18 \text{ years}$ (mean \pm standard deviation).” (page 10)

2. One of the main conclusions is that higher plasma A β 42/40 in African American participants was consistent with less amyloid pathology. However, no comparison was made to for example amyloid load on PET, no information on number (%) of amyloid-positive or amyloid-negative individuals, and no autopsy findings were available to support these findings. Furthermore, would it be possible that non-AD related factors are the cause of lower A β 40 levels, and as a result higher A β 42/40 as well? Would perhaps racial-specific cut-points be necessary to determine amyloid pathology? My advice would be to nuance the conclusion and add to the discussion on possible alternative explanations for these findings.

Thank you for this suggestion. Although the numbers are lower, we have now added this to Table 2 and our results: “Using the same covariate adjusted models, levels of amyloid burden by PET and CSF A β 42/40 were compared for Black and White groups (Table 2). Notably, the power to discern differences was lower because many fewer individuals underwent amyloid PET or CSF collection. The covariate-adjusted mean amyloid PET Centiloid values were slightly lower in 89 Black individuals compared to 626 White individuals, although this did not reach significance (21.6 ± 13.6 Centiloids versus 28.2 ± 13.4 Centiloids, $p=0.072$). The covariate-adjusted mean CSF A β 42/40 values were higher, consistent with lower amyloid burden, in 80 Black individuals compared to 806 White individuals (0.1122 ± 0.0068 versus 0.1069 ± 0.0068 , $p=0.014$). CSF A β 42 levels were slightly (8.2%) lower in Black individuals compared to White individuals (840 ± 117 pg/mL versus 915 ± 116 pg/mL, $p=0.041$), but CSF A β 40 levels were much (23.8%) lower in Black individuals than White individuals ($5,552 \pm 1,277$ pg/mL versus $7,284 \pm 1,272$ pg/mL, $p<0.0001$). CSF t-tau and p-tau181 levels were less abnormal in Black individuals compared to White individuals (t-tau: 114.2 ± 83.1 pg/mL versus 198.1 ± 82.9 pg/mL, $p=0.001$; p-tau181: 17.42 ± 12.31 pg/mL versus 28.46 ± 12.27 pg/mL, $p=0.004$).” (page 12)

Additionally, we addressed the use of race-specific cut-offs in our discussion, “There have been suggestions to decrease the level of amyloid biomarker abnormality required for Black individuals to increase enrollment of this under-represented group.

However, the use of race-specific cut-offs could have unintended consequences and even lead to systematic racial discrimination⁵⁶. For example, if Black individuals included in clinical trials had less amyloid pathology, they may be less responsive to amyloid-lowering treatments, potentially leading to the erroneous conclusion that the treatments are not effective in Black individuals. These concerns provide a strong rationale for not using race-specific cut-offs, even if the intent is to increase representation of Black individuals in AD clinical trials. (page 16)

3. While the main focus of the paper is longitudinal plasma biomarkers, this topic was slightly underrepresented in the results section. The first paragraph of the results (cohort characteristics) could be shortened, while paragraph 3.4 could be expanded: a. I was surprised to see that A β 42 increased in Black (and White?) participants, while I would expect these levels to decrease. Do you have an explanation for these findings? And does A β 42/40 increase or decrease in each group?

Part of the reason the longitudinal results are underrepresented is the negative findings in racial differences. However, to describe this in more details we have expanded the results from longitudinal analyses by including the estimated rates of change and the SE for each racialized group. Additionally, we have now reported more analyses on the possible modifying effects of main AD risk factors and comorbidities on the longitudinal racial differences in newly added Supplementary Tables 5-8. We were also surprised to see plasma A β 42 increased in Black (and White) participants longitudinally, but we observed that plasma A β 42/40 does decrease in both groups longitudinally.

We have added discussions of racial differences related to A β 40 and A β 42: “Interestingly, the higher average plasma A β 42/40 levels in Black participants resulted from lower levels of plasma A β 40 but similar levels of plasma A β 42 compared to White participants. Aligned with this finding, CSF A β 40 levels were much lower in Black participants compared to White participants. Importantly, it is the ratio of A β 42 to A β 40 that reflects sequestration of A β 42 into amyloid plaques and is most strongly associated with amyloid plaque burden⁵⁷. Recent work has found that individuals have significant differences in overall levels of CSF proteins that are driven by non-AD related factors, and that CSF A β 40 is a useful measure of an individual’s overall CSF protein level⁵⁸⁻⁶⁰. The reasons for differences in overall levels of CSF proteins are not yet well understood, but may be related to physiological factors in CSF production and clearance that are associated with age, sex, ventricular volumes, and circadian rhythms^{59,61}. Additional factors may influence the proportion of central nervous system-derived proteins that reach the plasma⁶². Normalizing fluid biomarkers of AD (e.g., CSF p-tau181) to A β 40 may decrease non-AD related variance and improve associations with AD pathology^{58,60}. Our findings suggest that Black individuals have lower levels of CSF and plasma A β 40. It is possible that normalization to A β 40 may reduce racial differences in biomarkers of AD pathology^{16,28}” (page 16-17)

b. It would help interpret these findings if you could include a table with the results of the linear mixed models (including δ (SE) estimates for biomarker slopes for both Black and White participants), and perhaps a figure as well.

We have now added this as Table 3 and newly added Supplementary Figure 2.

c. In CSF, A β levels are known to plateau once amyloid becomes abnormal, and the rate of change differs depending on amyloid status. I would be interested in seeing a subgroup-analysis of longitudinal changes in plasma A β ₄₂, A β ₄₀ and A β _{42/40} levels in A- vs A+ individuals (based on CSF or PET) in both African American and White participants.

Thanks for this suggestion. We have now added analyses stratified by CSF or amyloid PET status for baseline plasma A β levels in newly added Supplementary Table 1 and longitudinal rate of change for plasma A β biomarkers in newly added Supplementary Table 5. There was no effect of amyloid status on racial differences in baseline plasma A β levels or longitudinal rate of change for plasma A β biomarkers.

d. Furthermore, for those who have CSF and PET available: how many participants were A- and A+?

These numbers are now included in newly added Supplementary Table 1. Given concerns about whether cut-offs are appropriate in all groups, we have intentionally emphasized continuous levels for amyloid PET and the CSF biomarkers for Black and White individuals (Table 2).

4. Last, the influence of most covariates on baseline and longitudinal A β levels in both races was examined, except for the influence of hypertension and diabetes mellitus. Why did the authors chose to exclude those factors from these analyses? I would be interested in seeing the results, especially as hypertension and diabetes are a risk factor for chronic kidney disease, which is known to influence plasma biomarker levels. Also: where measures of kidney functioning available?

We have now added the effects of hypertension and diabetes on baseline A β in Supplementary Tables 2-4 and longitudinal rate of change of A β in Supplementary Table 6-8. Unfortunately our database did not contain data on kidney function or chronic kidney disease, and we have included this as another limitation of our study (page 15).

Reviewer #3 (Remarks to the Author):

1) The authors might speculate as to why the Black participants might have a lower baseline A β ₄₀ level.

We have added additional discussion regarding this important point: “Interestingly, the higher average plasma A β _{42/40} levels in Black participants resulted from lower levels of plasma A β ₄₀ but similar levels of plasma A β ₄₂ compared to White participants. Aligned with this finding, CSF A β ₄₀ levels were much lower in Black participants compared to White participants. Importantly, it is the ratio of A β ₄₂ to A β ₄₀ that reflects sequestration of A β ₄₂ into amyloid plaques and is most strongly associated with amyloid plaque burden⁵⁷. Recent work has found that individuals have significant differences in

overall levels of CSF proteins that are driven by non-AD related factors, and that CSF A β 40 is a useful measure of an individual's overall CSF protein level⁵⁸⁻⁶⁰. The reasons for differences in overall levels of CSF proteins are not yet well understood, but may be related to physiological factors in CSF production and clearance that are associated with age, sex, ventricular volumes, and circadian rhythms^{59,61}. Additional factors may influence the proportion of central nervous system-derived proteins that reach the plasma⁶². Normalizing fluid biomarkers of AD (e.g., CSF p-tau181) to A β 40 may decrease non-AD related variance and improve associations with AD pathology^{58,60}. Our findings suggest that Black individuals have lower levels of CSF and plasma A β 40. It is possible that normalization to A β 40 may reduce racial differences in biomarkers of AD pathology^{16,28}." (page 16-17)

2) There should be clarification that Black and White participants across the sites represent a random sample from the community. Selection factors and/or inclusion/exclusion criteria must be similar across centers and no temporal differences exist between the ascertainment of measures between Whites and Blacks, to ensure that these are not confounding factors.

We addressed these issues in our discussion: "Limitations of our study include that AD research cohorts are not representative of the general population and do not represent a random sample from the community" (page 18), and "The cohort included a relatively large number of Black participants from three sites that have similar inclusion and exclusion criteria and use a uniform clinical assessment protocol³⁸. Plasma samples were collected according to standard protocols that did not vary by racial group and samples were assayed together at a single location." (page 17)

REVIEWERS' COMMENTS

Reviewer #2 (Remarks to the Author):

This study provides important and unique data on plasma Ab42/40 levels in a biracial cohort. The authors have addressed my comments and those of Reviewer #1 to my satisfaction. I believe the additional analyses and remarks have contributed to the quality of this manuscript. As such, I recommend this article for publication.

Reviewer #3 (Remarks to the Author):

The authors have improved their paper with the inclusion of additional data and more detailed discussion.